# Overexpression of Ssd1 and calorie restriction extend yeast replicative lifespan by preventing deleterious age-dependent iron uptake

J Ignacio Gutierrez[1]*, Claudia Edgar[1,2], Jessica K Tyler[1]*

[1]Department of Pathology and Laboratory Medicine, Weill Cornell Medicine, New York, United States; [2]BCMB Graduate Program, Weill Cornell Graduate School of Biomedical Sciences, New York, United States

## eLife Assessment

This **important** study uses innovative microfluidics-based single-cell imaging to monitor replicative lifespan, protein localization, and intracellular iron levels in aging yeast cells. The evidence for the proposed role of Ssd1 and reduced nutrients for lifespan through limiting iron uptake is **convincing**, even though some mechanistic details remain unclear. This work will be of interest to cell biologists working on aging and iron metabolism.

*For correspondence:
nachobop@gmail.com (JIG);
jet2021@med.cornell.edu (JKT)

Competing interest: The authors declare that no competing interests exist.

**Abstract** Overexpression of the mRNA binding protein Ssd1 extends the yeast replicative lifespan. Using microfluidics to trap and image single cells throughout their lifespans, we find that lifespan extension by Ssd1 overexpression is accompanied by formation of cytoplasmic Ssd1 foci. The age-dependent Ssd1 foci are condensates that appear dynamically in a cell-cycle-dependent manner, and their failure to resolve during mitosis coincided with the end of lifespan. Ssd1 overexpression was epistatic with calorie restriction (CR) for lifespan extension, and yeast overexpressing Ssd1 or undergoing CR were resistant to iron supplementation-induced lifespan shortening, while their lifespans were reduced by iron chelation. The nuclear translocation of the Aft1 transcriptional regulator of the iron regulon occurred during aging in a manner that predicted remaining lifespan but was prevented by CR. Accordingly, age-dependent induction of the Fit2 and Arn1 high-affinity iron transporters within the iron regulon was reduced by CR and Ssd1 overexpression. Consistent with age-dependent activation of the iron regulon, intracellular iron accumulated during aging but was prevented by CR and Ssd1 overexpression. Moreover, lifespan extension by Ssd1 overexpression or CR was epistatic to inactivation of the iron regulon. These studies reveal that CR and Ssd1 overexpression extend the yeast replicative lifespan by blocking deleterious age-dependent iron uptake, identifying novel therapeutic targets for lifespan extension and providing insight into how CR may extend the lifespan and healthspan in humans.

## Introduction

Aging is a gradual and irreversible biological process that results in the progressive decline in physical function of cells and tissues. The fundamental processes that go awry during aging are conserved across eukaryotes, exemplified by the fact that anti-aging regimens such as calorie restriction (CR) and mTOR/Tor1 pathway inhibition using rapamycin function to extend lifespan in species ranging from budding yeast to mammals (*Anderson and Weindruch, 2010*; *Blagosklonny, 2019*). There is a great

deal of interest in the identification of common features, or hallmarks, of aging (*López-Otín et al., 2023*) and in determining which of these hallmarks are reversed by anti-aging regimens.

Loss of cellular homeostasis is a hallmark of aging, characterized by the gradual decline in the ability to regulate intracellular conditions leading to accumulation of cellular damage, impaired stress responses, and functional deterioration across tissues (*López-Otín et al., 2023*; *López-Otín et al., 2013*). Age-associated dysregulation of transcription and translation may contribute to this loss of homeostasis during aging as it can cause imbalanced protein production, impaired stress responses, and metabolic disruption leading to accumulation of damaged proteins (*Anisimova et al., 2020*; *Hu et al., 2018*). In this context, Ssd1, a yeast mRNA-binding protein involved in modulating the stability, localization, or translation of specific mRNAs (*Luukkonen and Séraphin, 1999*), may play a pivotal role in preserving cellular homeostasis during aging given that its deletion shortens the yeast replicative lifespan (*Kaeberlein et al., 2004*) while its overexpression extends lifespan (*Hu et al., 2018*). Yeast is the only type of eukaryotic cell where the length of the replicative lifespan (RLS), the number of times that a cell divides before death, can be measured accurately (*Steinkraus et al., 2008*). Previously, we have shown that overexpression of Ssd1 extends yeast RLS by suppressing global translation efficiency (*Hu et al., 2018*) possibly involving its stress-induced role in mRNA sequestration into cytoplasmic mRNA storage granules termed P-bodies and stress granules where the mRNAs can be stored or degraded (*Kurischko et al., 2011*). Ssd1 has been shown to bind a large number of mRNAs, with some reports identifying up to 300 targets (*Hogan et al., 2008*; *Dutcher and Gasch, 2024*). However, whether sequestration of a subset of these mRNAs by overexpressed Ssd1 is sufficient for lifespan extension remains unclear. Additionally, the identity of the mRNAs whose translation is suppressed by overexpressed Ssd1 to extend lifespan is unknown.

By contrast to yeast lifespan extension by Ssd1 overexpression, CR has been shown to extend lifespan in many organisms including primates (*Caristia et al., 2020*; *Colman et al., 2014*). However, CR reprograms metabolism affecting a large portion of cellular processes (*Anderson and Weindruch, 2010*), making it difficult to determine the precise molecular mechanism of CR-mediated lifespan extension. One putative mechanism for improved cellular homeostasis by CR leading to lifespan extension implicates CR's ability to reduce reactive oxygen species (ROS) production, thereby decreasing oxidative stress (*Ungvari et al., 2008*). Studies in yeast have suggested that CR reduces ROS levels by mechanisms including activating superoxide dismutases (*Mesquita et al., 2010*) as well as by improving mitochondrial function (*Hughes and Gottschling, 2012*; *Choi and Lee, 2013*). Another mode for reduction of oxidative stress by CR, at least in the mouse heart, is decreased cellular iron uptake to improve iron homeostasis (*An et al., 2020*). In this situation, mechanistically, CR resulted in the downregulation of transcription of genes encoding iron transporters that mediate iron import and upregulation of ferritin to sequester iron (*An et al., 2020*). Whether CR downregulates iron uptake to lead to lifespan extension is an open question.

Iron is an essential cofactor for numerous enzymatic processes, being required for the proper function of enzymes involved in oxygen transport, electron transfer, DNA replication and repair, and various metabolic pathways (*Ben Zichri-David et al., 2025*). However, in its free form (free labile iron, $Fe^{+2}$), iron is highly reactive and induces the production of ROS, thereby generating oxidative stress (*Chutvanichkul et al., 2018*). For this reason, iron uptake is a tightly regulated process such that when the levels of cellular iron are sufficient, iron transport into the yeast cells is mediated by only low-affinity iron transporters (*Martínez-Garay et al., 2016*). When iron levels in the cell are deficient, the transcription factors Aft1 and Aft2 translocate from the cytoplasm to the nucleus in yeast *Yamaguchi-Iwai et al., 2002* to induce the iron regulon: a set of genes encoding machinery that coordinates iron uptake (including high-affinity iron transporters and transporters of siderophores that scavenge iron from the extracellular environment), vacuolar mobilization, and metabolic adaptation to maintain cellular iron homeostasis (*Ramos-Alonso et al., 2020*; *Rutherford et al., 2005*). Recent reports indicate that the activation of the iron regulon occurs during yeast replicative aging as a result of reduced levels of mitochondrial iron-sulfur clusters (*Chen et al., 2020b*; *Patnaik et al., 2022*). We hypothesized that age-dependent activation of the iron regulon would lead to accumulation of free labile iron in cells during aging that would be detrimental to lifespan.

Here, we sought to identify the molecular mechanisms whereby Ssd1 overexpression and CR extend the yeast RLS. Using fluorescent imaging of single cells immobilized in microfluidic devices throughout their lifespans, we find that lifespan extension by overexpression of Ssd1 is accompanied

by formation of cytoplasmic Ssd1 foci. These foci appear dynamically after DNA replication and resolve by cytokinesis, while their failure to resolve by cytokinesis coincides with the end of lifespan. Furthermore, Ssd1 overexpression is epistatic with CR for lifespan extension. Our results indicate that both CR and Ssd1 overexpression prevent the activation of the iron regulon during aging, which in turn prevents the increased uptake of iron that usually occurs during aging. Furthermore, we show that derailing age-dependent activation of the iron regulon is the molecular mechanism whereby CR and Ssd1 extend replicative lifespan.

## Results

### Lifespan extension by overexpression of Ssd1 is accompanied by age-dependent appearance of Ssd1 condensates

The Ssd1 mRNA-binding protein has a prion-like domain (*Kurischko and Broach, 2017*) which has been shown to be involved in the localization of overexpressed Ssd1 to cytoplasmic condensates including P-bodies and stress granules in response to glucose starvation (*Kurischko and Broach, 2017*). Because the conditions in which Ssd1 overexpression extends the yeast lifespan are glucose replete (*Hu et al., 2018*), we asked whether Ssd1 even forms foci/condensates during replicative aging. Therefore, we generated a yeast strain overexpressing eGFP-tagged Ssd1 by integrating *SSD1*-eGFP under the control of a strong *GPD1* promoter (P$_{GDP1}$-SSD1-GFP). This promoter has been used extensively to overexpress *SSD1* in previous studies (*Kurischko et al., 2011*; *Kurischko and Broach, 2017*). We also made a strain expressing eGFP-tagged Ssd1 from the endogenous *SSD1* promoter (SSD1-GFP). We confirmed by flow cytometry that overexpression of Ssd1 occurred in seemingly all P$_{GDP1}$-SSD1-GFP cells (*Figure 1A*). We then measured the RLS of these strains on microfluidic devices (*Gutierrez and Tyler, 2024*) and observed a robust lifespan extension in the strain overexpressing *SSD1* (P$_{GPD1}$-SSD1-GFP; *Figure 1B*). By contrast, a strain deleted for *SSD1* had a shortened RLS (*Figure 1B*), as published previously (*Kaeberlein et al., 2004*).

We next asked whether overexpression of Ssd1 leads to formation of Ssd1 foci during replicative aging. Using microfluidics combined with fluorescence microscopy (*Gutierrez and Tyler, 2024*), we tracked single cells from the SSD1-GFP (control) or P$_{GDP1}$-SSD1-GFP strains throughout their entire lifespan, acquiring images every 20 min over approximately 4 days. We observed that in P$_{GPD1}$-SSD1-GFP cells, but not in the control strain, transient Ssd1 foci appeared during aging (*Figure 1C and D*, *Figure 1—figure supplement 1*, *Figure 1—videos 1–4*). The abundance of the Ssd1 protein increased over the first 4–6 cell divisions of the RLS before reaching a plateau, both in control and P$_{GPD1}$-SSD1-GFP cells (*Figure 1D*), where the overall fluorescence intensity in P$_{GPD1}$-SSD1-GFP cells was approximately tenfold higher than in the control strain (*Figure 1D*, *Figure 1—figure supplement 2*). No Ssd1-GFP foci were observed in P$_{GPD1}$-SSD1-GFP cells during the early cell divisions in the lifespan (*Figure 1D*), suggesting that either a threshold concentration of Ssd1 may be required for foci formation and/or that alterations in cell homeostasis during aging promote Ssd1 foci formation in cells overexpressing Ssd1. During aging, Ssd1 foci appeared in all cells overexpressing Ssd1, and the number and frequency of Ssd1 foci appearance varied between cells, with the highest number of foci per cell apparent at the end of the lifespan (*Figure 1D*). These data show that the lifespan extension that results from overexpression of Ssd1 is accompanied by age-dependent formation of Ssd1 foci, suggesting that the ability to form Ssd1 foci correlates with increased longevity while the appearance of five or more Ssd1 foci may be either a cause or consequence of end of lifespan.

Similar to the starvation-induced Ssd1-GFP foci in Ssd1 overexpressing cells (*Kurischko and Broach, 2017*), age-induced Ssd1-GFP foci in Ssd1 overexpressing cells are also cytoplasmic (*Figure 1—figure supplement 3A*). Moreover, Ssd1-GFP foci can be dissolved by 1,6-hexanediol, suggesting that they are condensates (*Figure 1—figure supplement 3B*). Furthermore, age-induced Ssd1-GFP foci underwent fusion and fission (*Figure 1—video 5*), consistent with them also being condensates.

### Ssd1 condensates form in cells overexpressing Ssd1 during aging in a cell-cycle-regulated manner and are distinct from age-induced P-bodies and Hsp104 foci

To investigate further the transient nature of appearance of the Ssd1 condensates during aging (*Figure 1C and D*), we used bud size as a proxy for cell cycle stage (*Figure 2A*). We found that

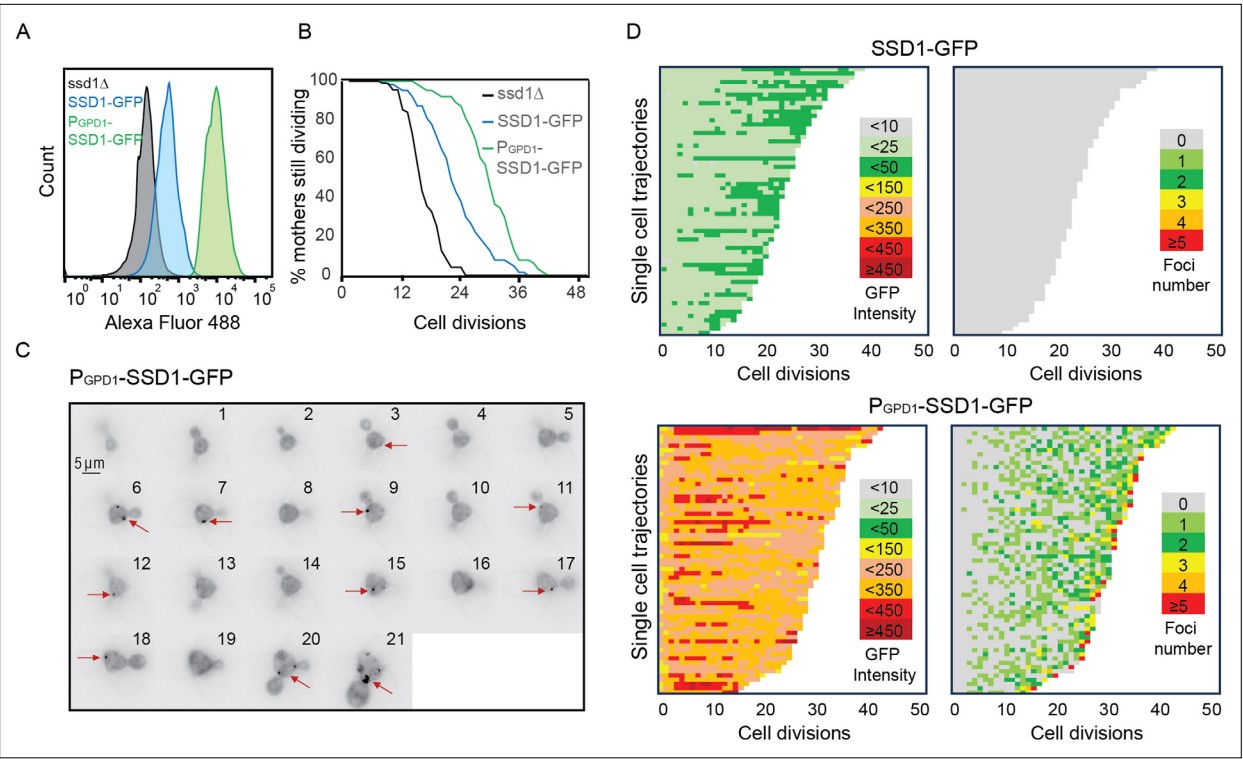

**Figure 1.** Overexpression of Ssd1 extends lifespan and forms transient cytoplasmic foci during replicative aging. (**A**) Levels of GFP tagged Ssd1 in the indicated strains, for 10,000 cells per strain. (**B**) RLS of strains shown in panel A for 90 $ssd1\Delta$ cells, 63 SSD1-GFP cells and 63 $P_{GPD1}$-SSD1-GFP cells, $p$-values are as follows: for control strain versus $ssd1\Delta$ strain is 0, for control versus $P_{GPD1}$-SSD1-GFP is $8\times10^{-8}$. (**C**) Transient formation of Ssd1-GFP foci in $P_{GPD1}$-SSD1-GFP strain during aging at the indicated cell division number of the RLS for one cell. Arrows indicate Ssd1-GFP foci. (**D**) Single cell trajectories showing total cell Ssd1-GFP intensity and number of Ssd1-GFP foci formation for the indicated strains throughout the RLS. Each rectangle represents one cell cycle.

The online version of this article includes the following video, source data, and figure supplement(s) for figure 1:

**Source data 1.** Replicative lifespan data from the microfluidic imaging used to generate the graphs.

**Figure supplement 1.** SSD1 foci formation in a single cell during replicative aging.

**Figure supplement 2.** Relative Ssd1-GFP protein amount per cell throughout the lifespan.

**Figure supplement 3.** Properties of the Ssd1 foci.

**Figure 1—video 1.** Bright field and corresponding fluorescence microscopy.
https://elifesciences.org/articles/108892/figures#fig1video1

**Figure 1—video 2.** Bright field of Ssd1-GFP for an SSD1-GFP cell during the RLS.
https://elifesciences.org/articles/108892/figures#fig1video2

**Figure 1—video 3.** Bright field and corresponding fluorescence microscopy.
https://elifesciences.org/articles/108892/figures#fig1video3

**Figure 1—video 4.** Bright field of Ssd1-GFP for a $P_{GPD1}$-SSD1-GFP cell during the RLS.
https://elifesciences.org/articles/108892/figures#fig1video4

**Figure 1—video 5.** Fluorescence microscopy of Ssd1-GFP condensates undergoing fusion in an aging $P_{GPD1}$-SSD1-GFP cell.
https://elifesciences.org/articles/108892/figures#fig1video5

Ssd1 condensates typically appeared in cells overexpressing Ssd1 during aging in the $G_2$/M cell cycle stages, when the bud reaches its maximum size, and very rarely during $G_1$/S, when the bud was absent or small (*Figure 2A*). We also observed that Ssd1 condensates virtually always lasted less than one cell cycle, dissolving before or during cytokinesis of the same cell cycle in which they formed (*Figure 2A*). However, the Ssd1 condensates were not dissolved before mitosis in some cells in their last division of their RLS (*Figure 2A*). As such, it appears that the ability to both form and resolve Ssd1 foci during

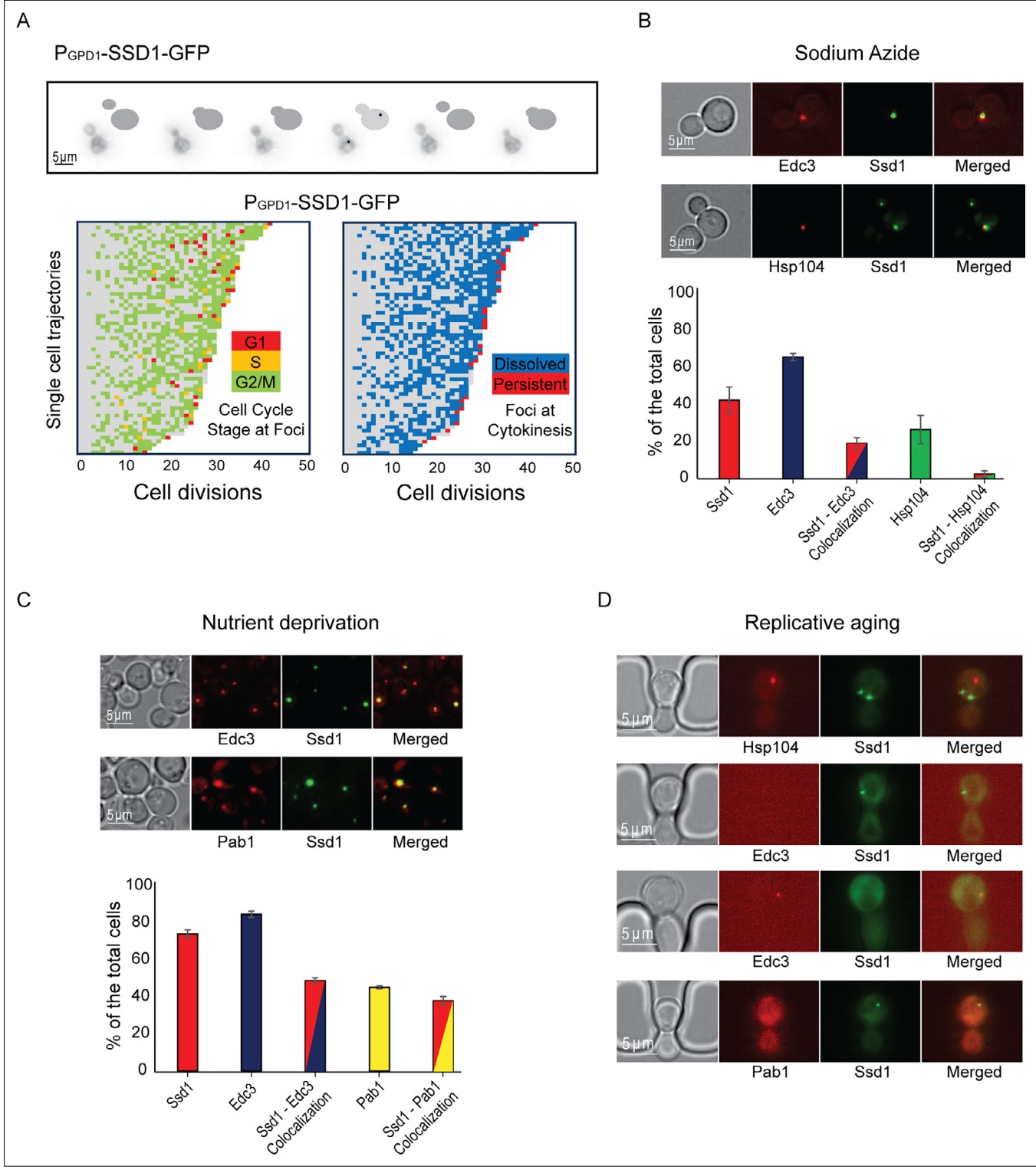

**Figure 2.** Ssd1 foci formed during aging are cell cycle regulated and distinct from Hsp104 foci, P-bodies, and stress granules. (**A**) Example showing Ssd1-GFP foci in $G_2$ phase during aging. The images are taken of the same cell every 20 min and the cartoons indicate the bud size. Below are shown single cell trajectories of cell cycle stage at which Ssd1 foci are present during aging (left) and whether the foci dissolved at the following cytokinesis (shown in blue) or were persistent through cytokinesis (shown in red; right). (**B**) Colocalization of Ssd1 foci with P-bodies and Hsp104 foci in cells treated with 10 mM Sodium azide. Quantitation is shown below for over 100 cells per condition. (**C**) Colocalization of Ssd1 foci with P-bodies (marked by Edc3) and stress granules (marked by Pab1) under nutrient deprivation and quantitation. (**D**) Failure of Ssd1 foci to colocalize with age-induced Hsp104 foci or P-bodies.

*Figure 2 continued on next page*

*Figure 2 continued*

The online version of this article includes the following source data and figure supplement(s) for figure 2:

**Source data 1.** Replicative lifespan data from the microfluidic imaging used to generate the graphs.

**Figure supplement 1.** Frequency of Ssd1 foci formation.

aging correlates with lifespan extension, while failure to resolve Ssd1 foci correlates with the end of lifespan in some cells.

We were curious to investigate the relationship of the Ssd1 condensates that occur in cells over-expressing Ssd1 during aging to other cytoplasmic condensates that Ssd1 has been reported to colocalize with, when overexpressed. Ssd1 foci formation has primarily been studied in the context of the stress response (*Kurischko and Broach, 2017*), where overexpressed Ssd1 colocalizes with stress granules and P-bodies upon glucose depletion, hypertonic stress, and heat shock, for trans-lational repression by sequestering untranslated mRNAs (*Kurischko et al., 2011*), or helping miti-gate proteotoxic stress by directing misfolded proteins to IPODs (insoluble protein deposit sites; *Kurischko and Broach, 2017*; *Hose et al., 2020*). While the strain overexpressing Ssd1 ($P_{GPD1}$-SSD1) only very rarely exhibited Ssd1 foci during exponential growth (*Figure 2—figure supplement 1*), we observed a significant increase in Ssd1 foci formation under stress: approximately 40% and 75% of $P_{GPD1}$-SSD1 cells displayed Ssd1 foci following treatment with 10 mM sodium azide or growth to saturation (nutrient deprivation), respectively (*Figure 2B and C*). Sodium azide, which inhibits the electron transport chain, induced P-body formation in ~65% of cells, as marked by mCherry-tagged Edc3. Approximately 20% of cells showed Ssd1 localized to P-bodies in response to sodium azide treatment (*Figure 2B*). Similarly, tagging Hsp104 with mCherry revealed that upon sodium azide treat-ment, ~25% of cells showed Hsp104 foci and 5% of cells had Hsp104 colocalizing with Ssd1 foci (*Figure 2B*). Under nutrient deprivation, ~85% of cells formed P-bodies and ~50% of cells showed colocalization with Ssd1 foci (*Figure 2C*). In the same condition, ~45% of cells exhibited stress gran-ules, as indicated by mCherry-tagged Pab1, and ~40% of cells also displayed colocalization with Ssd1 foci (*Figure 2C*). These results indicate that in cells overexpressing Ssd1, Ssd1 frequently but not always colocalized with stress-induced foci under acute stress conditions.

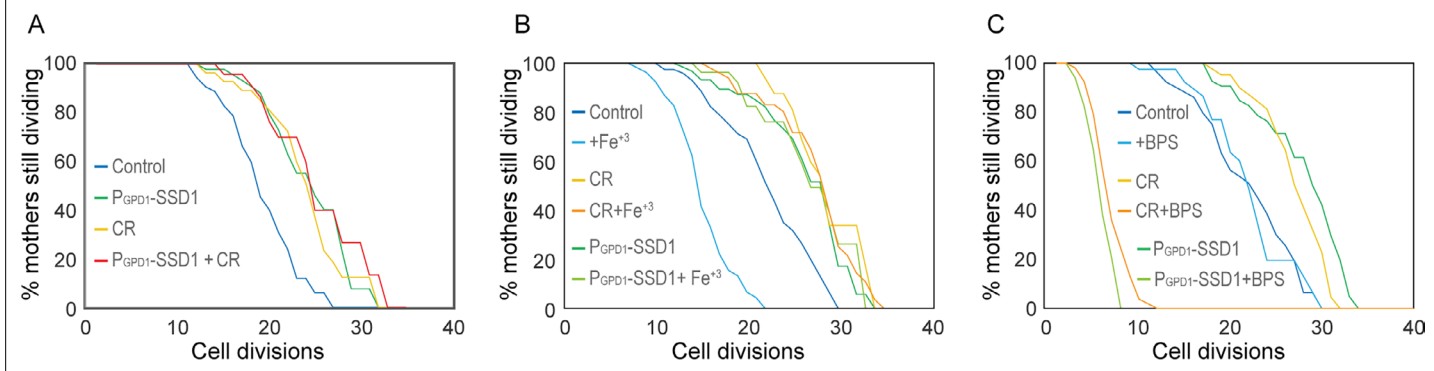

**Figure 3.** CR and overexpression of Ssd1 are epistatic for lifespan extension, both influencing iron metabolism. (**A**) Calorie restriction and overexpression of Ssd1 are epistatic for extension of RLS, for 54 control cells, 48 $P_{GPD}$-SSD1 cells, 48 control cells undergoing CR, and 44 $P_{GPD}$-SSD1 cells + CR. p-values determined by Student's T-test for control versus $P_{GPD}$-SSD1, CR, and $P_{GPD}$-SSD1+CR are $1\times10^{-6}$, $3\times10^{-4}$ and $5\times10^{-4}$ respectively. There is no significant difference between $P_{GPD}$-SSD1, CR, and $P_{GPD}$-SSD1+CR. (**B**) CR and overexpression of Ssd1 protect cells from lifespan reduction due to iron supplementation with 10 µM $Fe^{+3}$, for 55 control cells, 53 control cells + $Fe^{+3}$, 47 $P_{GPD}$-SSD1 cells, 41 $P_{GPD}$-SSD1 + $Fe^{+3}$ cells, 37 CR cells, and 50 CR + $Fe^{+3}$ cells. p-value for control versus $Fe^{+3}$ is $6\times10^{-7}$. There is no significant difference between $P_{GPD}$-SSD1 with and without $Fe^{+3}$ or for CR with and without $Fe^{+3}$. (**C**) CR and Ssd1 overexpression protect cells from the lifespan reduction caused by iron depletion with 200 µM BPS, for 52 control cells, 40 control cells + BPS, 60 $P_{GPD}$-SSD1 cells, 34 $P_{GPD}$-SSD1 cells + BPS, 44 control cells under CR, and 53 cells under CR + BPS. p-values for the control + BPS, versus $P_{GPD}$-SSD1+BPS and CR + BPS are 0, while there is no significant difference between the control with and without BPS treated and untreated control are not significantly different; control compared to $P_{GPD}$-SSD1 and CR is 0.

The online version of this article includes the following source data for figure 3:

**Source data 1.** Replicative lifespan data from the microfluidic imaging used to generate the graphs.

To determine whether Ssd1 foci that form during aging also colocalize with P-bodies, stress granules, or Hsp104 foci, we examined these condensates in aging $P_{GPD1}$-SSD1 cells. Hsp104 foci have been shown to form around the fifth cell division of the RLS and are asymmetrically retained in mother cells throughout the lifespan (*Saarikangas and Barral, 2015*). Consistent with this, we observed Hsp104 foci during aging, but we never observed them colocalized with Ssd1 condensates (*Figure 2D*) even though Ssd1-GFP condensates form in every cell during aging (*Figure 1D*). Previous reports indicate that P-bodies appear in late lifespan (*Hu et al., 2018*); we similarly observed Edc3-mCherry foci (marking P-bodies) in late lifespan, but again, we did not observe them ever colocalizing with Ssd1 foci (*Figure 2D*). We did not observe the formation of stress granules as determined by Pab1-mCherry staining during the RLS (*Figure 2D*). These data indicate that the Ssd1 condensates that form during aging in cells overexpressing Ssd1 are distinct from P-bodies, stress granules, and Hsp104 foci.

## Overexpression of Ssd1 is epistatic to calorie restriction for lifespan extension, and both reduce activation of the iron regulon during aging

To gain mechanistic insight into how overexpression of Ssd1 and Ssd1 condensate formation may extend lifespan, we tested whether overexpression of Ssd1 ($P_{GPD1}$-SSD1) was epistatic to CR for lifespan extension. We found that the RLS extension resulting from Ssd1 overexpression was equivalent to that achieved by CR (0.05% glucose) and moreover, was equivalent to the RLS of cells that overexpressed Ssd1 during CR (*Figure 3A*). These results are consistent with Ssd1 overexpression and CR extending lifespan via a shared pathway.

Other studies in the lab on the effect of CR on iron metabolism had led us to be interested in the observation that the iron regulon is activated during aging (*Chen et al., 2020b*; *Patnaik et al., 2022*), presumably resulting in increased iron uptake in aged cells, which is likely to be deleterious for lifespan given that constitutive activation of the iron regulon shortens RLS (*Rutherford et al., 2005*). As such, we were curious whether overexpression of Ssd1 or CR may lead to a muted iron response during aging. If this were the case, we would expect the lifespan of cells overexpressing Ssd1 or undergoing CR to be less sensitive to iron supplementation. Indeed, the control strain had a significantly reduced RLS upon supplementation with 10 µM iron(III) chloride ($FeCl_3$), while the extended RLS of $P_{GPD1}$-SSD1 and CR-treated cells was unaffected by 10 µM $FeCl_3$ (*Figure 3B*). To further probe iron dependency, we supplemented the media with 200 µM of the iron chelator bathophenanthroline disulfonic acid (BPS; *Prasad et al., 2006*). The RLS of control cells was unaffected by BPS, whereas $P_{GPD1}$-SSD1 and CR-treated cells exhibited a dramatically shortened lifespan (*Figure 3C*). These data demonstrate that yeast overexpressing Ssd1 or undergoing CR extend RLS by a shared pathway that is resistant to iron supplementation but sensitive to iron chelation.

## Overexpression of Ssd1 and calorie restriction derail the iron regulon during aging

Given that the lifespan extension caused by overexpressing Ssd1 and CR was not prevented by iron supplementation, which shortened the lifespan of control cells (*Figure 3B*), we sought to determine whether CR or Ssd1 overexpression disrupted age-dependent activation of the iron regulon, which could theoretically prevent deleterious iron uptake. Under low-iron conditions, Aft1 translocates to the nucleus to activate transcription of the iron regulon (*Yamaguchi-Iwai et al., 2002*; *Figure 4A*, *Figure 4—figure supplement 1A*). We asked whether Aft1 localizes to the nucleus during aging and whether this is prevented in yeast overexpressing Ssd1 or during CR. Due to its low endogenous expression, we overexpressed a GFP-tagged version of *AFT1* from the GPD1 promoter, as previously reported (*Ueta et al., 2003*), to more effectively detect its cellular location. In control cells, we observed nuclear localization of GFP-Aft1 during mid-to-late lifespan (*Figure 4B*). Furthermore, there is a clear trend apparent where cells in which Aft1 translocated to the nucleus later in their lifespan tended to live longer (*Figure 4C*). Also, cells lived on average 5 more divisions after Aft1 translocated into the nucleus (*Figure 4—figure supplement 1B*), suggesting that Aft1 nuclear translocation may predict the end of lifespan. By contrast, CR-treated cells showed little or no nuclear localization of Aft1, except in some cells where it occurred very late in lifespan (*Figure 4B*). Notably, this is not due to reduced levels of Aft1 during CR (*Figure 4—figure supplement 1C*). These data indicate that the iron regulon is activated during aging and is predictive of the end of lifespan, while cells undergoing CR do not activate the iron regulon during aging.

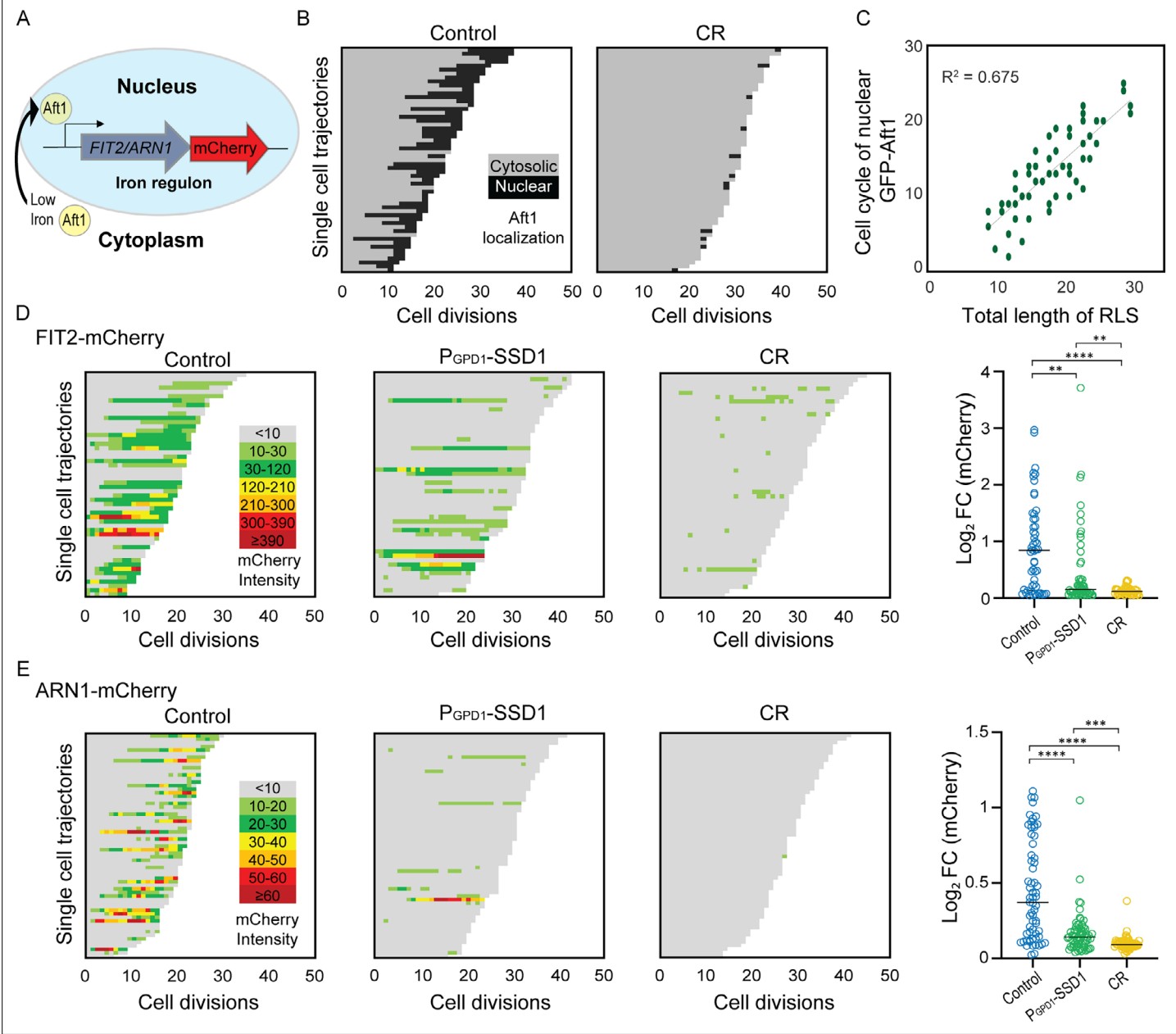

**Figure 4.** Age-dependent induction of the iron regulon predicts the end of lifespan and is blocked by Ssd1 overexpression and CR. (**A**) Schematic of iron regulon activation. (**B**) Single cell trajectories of nuclear localization of GFP-Aft1 during the RLS of control or CR yeast. (**C**) Earlier nuclear localization of Aft1 in the lifespan correlates with a shorter total lifespan. Analysis of the data shown in panel B. (**D**) Single cell trajectories of FIT2-mCherry expression during aging and Peak log fold change (FC) in mCherry intensity throughout the lifespan. (**E**) Single cell trajectories of ARN1-mCherry expression during aging and fold change intensity throughout the lifespan. Statistical difference is indicated where n.s. is not significant change with p>0.05, * p≤0.05, ** p≤0.01, *** p≤0.001 and **** p≤0.0001.

The online version of this article includes the following source data and figure supplement(s) for figure 4:

**Source data 1.** Replicative lifespan data from the microfluidic imaging used to generate the graphs.

**Figure supplement 1.** Aft1 localization and expression during aging.

**Figure supplement 2.** Replicative lifespans of strains expressing FIT2-mCherry and ARN1-mCherry.

**Figure supplement 3.** Single-cell examples of experiments shown in *Figure 4D and E*, to show variability between different cells.

**Figure supplement 4.** Replicative lifespans with and without *cth2Δ*.

We were unable to examine Aft1 localization during aging in the strain overexpressing Ssd1 because the high-level expression of GFP-Aft1 in this strain caused synthetic sickness. Instead, we sought to examine whether Ssd1 overexpression or CR prevents the induction of gene products within the iron regulon during aging. We analyzed expression of *FIT2* and *ARN1*, tagged with mCherry, two previously used reporters of iron regulon activity (*Diab and Kane, 2013*; *Gil et al., 2017*). Since loss-of-function mutations in *FIT2* and *ARN1* have been shown to extend the RLS (*McCormick et al., 2015*; *Ölmez et al., 2023*; *Yu et al., 2021*), we first confirmed that tagging these genes did not impair function: both control and P$_{GPD1}$-SSD1 strains carrying either mCherry tag exhibited an RLS equivalent to the untagged proteins under glucose rich and CR conditions (*Figure 4—figure supplement 2*, *Figure 3A*). In control strains, expression of Fit2 and Arn1 varied across the population, but generally increased with age (*Figure 4D, E*, *Figure 4—figure supplement 3*). In the strains overexpressing Ssd1, fewer cells induced Fit2 and Arn1 during aging compared to control strains, and their overall expression levels were lower upon Ssd1 overexpression (*Figure 4D and E*). In CR-treated strains, Fit2 and Arn1 levels remained very low throughout the lifespan (*Figure 4D and E*). Quantification of maximum expression per cell confirmed significantly higher levels of Fit2 and Arn1 in the control strain compared to P$_{GPD1}$-SSD1 and CR-treated cells (*Figure 4D and E*). These results demonstrate that overexpression of Ssd1 and CR greatly reduces age-related activation of the iron regulon.

It has been reported that preventing activation of the iron regulon benefits longevity by reducing Cth2 expression (*CTH2* is within the iron regulon), an mRNA-binding protein that targets for degradation the mRNA of non-essential iron binding proteins (*Patnaik et al., 2022*). We deleted *CTH2,* and in accordance with the literature, it extended lifespan compared to control strains; however, to a much lower extent than overexpressing Ssd1 or CR (*Figure 4—figure supplement 4*). This result indicates that preventing activation of the iron regulon benefits longevity in a manner that is not only due to preventing Cth2 expression.

## Experimental activation of the iron regulon prevents CR or Ssd1 overexpression from extending lifespan

To directly test whether activation of the iron regulon is a negative regulator of longevity, we examined the RLS in strains with a pre-activated iron regulon. In our experiments with the iron chelator BPS, we observed that while P$_{GPD1}$-SSD1 and CR-treated strains exhibited shorter lifespans in media supplemented with BPS, those strains were still capable of overnight growth in the presence of BPS. We took advantage of this observation by growing strains carrying the *FIT2*-mCherry or *ARN1*-mCherry reporter in media supplemented with 200 µM BPS overnight. Flow cytometry analysis of these exponentially growing cultures confirmed strong induction of the iron regulon in the control strain, the strain overexpressing Ssd1, and during CR (*Figure 5A*). We then measured RLS in the iron regulon 'induced' populations by transferring them to fresh control or CR media without BPS for the RLS measurements. Only cells showing *FIT2*-mCherry expression (i.e. those with an induced iron regulon) were tracked and compared to their counterparts grown without prior BPS exposure. We found that activation of the iron regulon reduced the RLS of the strain overexpressing Ssd1 and undergoing CR to a level equivalent to the RLS of the control strain (*Figure 5B*). As such, prior activation of the iron regulon prevents CR and Ssd1 overexpression from extending lifespan. These results are consistent with a model where suppression of activation of the iron regulon that normally occurs during aging is necessary for lifespan extension by CR and overexpression of Ssd1.

## Intracellular iron accumulates during aging, but not in yeast overexpressing Ssd1 or during CR

Artemisinin (Art) is a malaria drug that becomes toxic upon reacting with intracellular iron (*Chen et al., 2020a*). We hypothesized that if control cells accumulated more iron during aging as a consequence of activation of the iron regulon, they would show higher sensitivity to Art than cells overexpressing Ssd1 or during CR, which do not activate the iron regulon during aging to the same degree. Indeed, at 20 and 50 µM Art, control cells had a significantly reduced RLS compared to untreated controls (*Figure 5C*). In contrast, 20 µM Art had no effect on the RLS during CR or overexpressing Ssd1, and 50 µM Art only caused a modest decrease (*Figure 5C*). These results are consistent with CR and Ssd1 overexpression preventing the accumulation of free iron during aging that occurs when the iron regulon gets induced during aging.

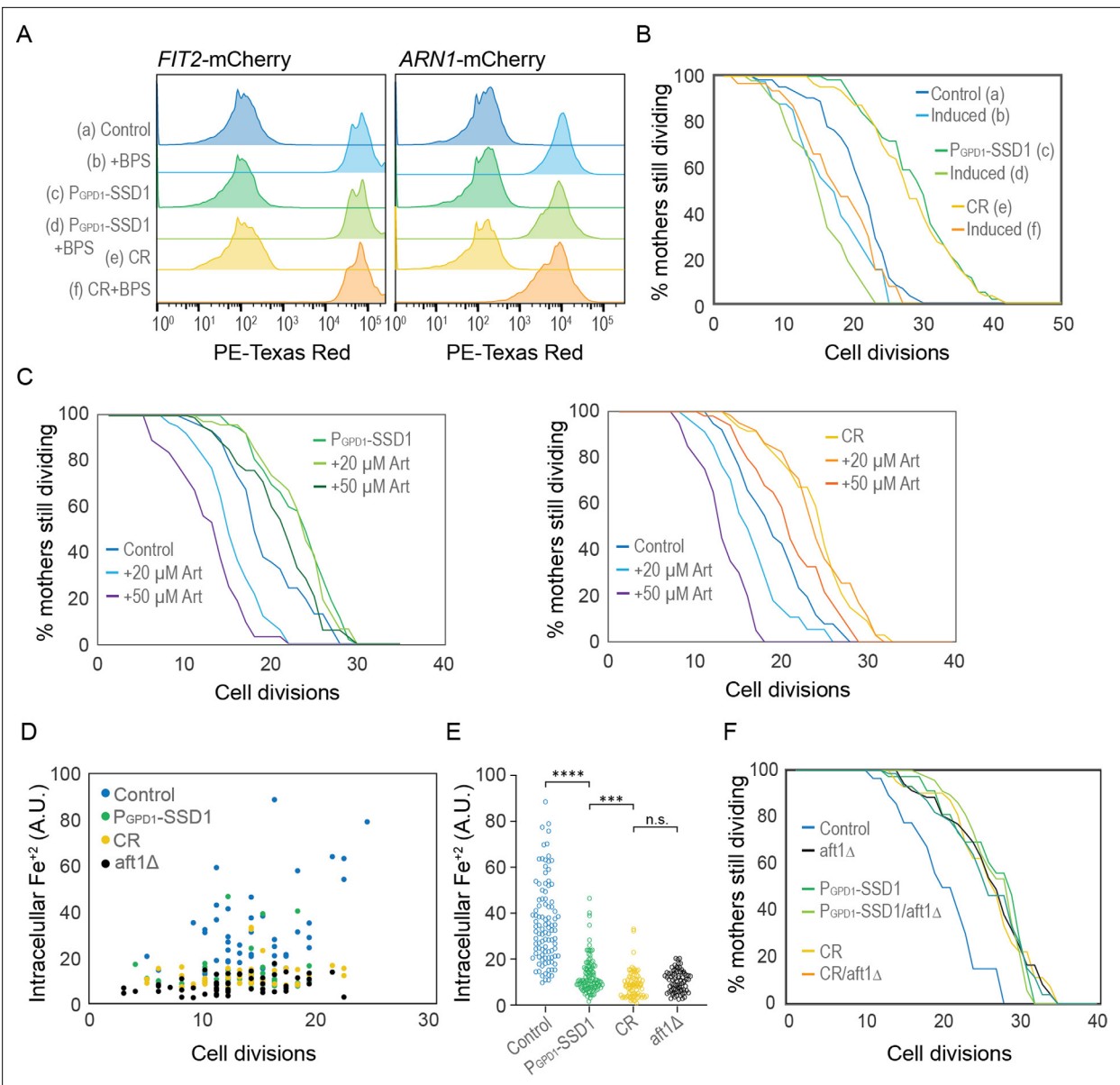

**Figure 5.** CR and Ssd1 overexpression extend lifespan by preventing the accumulation of iron resulting from age-dependent activation of the iron regulon. (**A**) Pre-induction of iron regulon by BPS treatment measured by mCherry-tagged expression of Fit2 and Arn1, for 10,000 cells per condition. (**B**) RLS of cells that had the iron regulon induced or uninduced from panel A measured by Fit2-mCherry expression, for 63 (**a**), 41 (**b**), 63 (**c**), 50 (**d**), 63 (**e**), and 31 (**f**) cells, p-values of control vs induced: 0.002, $P_{GPD1}$-SSD1 vs induced: 0 and CR vs induced: 0. (**C**) RLS of yeast overexpressing Ssd1 as well as under CR are protected from sensitivity to Artemisinin. Left plot: for 62 control, 64 20 µM Art, 37 50 µM Art, 86 $P_{GPD1}$-SSD1, 80 $P_{GPD1}$-SSD1+20 µM Art and 107 $P_{GPD1}$-SSD1+50 µM Art cells. p-values are control vs 20 µM Art: $4\times10^{-6}$, control vs 50 µM Art: 0, $P_{GPD1}$-SSD1 vs + $P_{GPD1}$-SSD1 20 µM Art: 0.98 and $P_{GPD1}$-SSD1 vs + $P_{GPD1}$-SSD1+50 µM Art: 0.01. Right plot: 38 control, 34 20 µM Art, 30 50 µM Art, 61 CR, 61 CR + 20 µM Art, and 50 CR + 50 µM Art cells. p-values are control vs 20 µM Art: 0.01, control vs 50 µM Art: $1\times10^{-7}$, CR vs CR 20 µM Art: 0.95 and CR vs+CR + 50 µM Art: 0.005. (**D**) Relative levels of $Fe^{2+}$ at indicated cell division of the RLS measured by Phen Green. (**E**) Relative amount of $Fe^{2+}$ in all cells of the indicated strains from panel D during the RLS, regardless of age. (**F**) RLS of the indicated strains / conditions with and without aft1Δ, for 39 control, 71 aft1Δ, 37 $P_{GPD1}$-SSD1, 72 $P_{GPD1}$-SSD1/aft1Δ, 42 CR, and 74 CR/aft1Δ cells. p-values of control compared to all other strains are $\leq 1 \times 10^{-4}$ while there is no significant difference between the strains / conditions other than the control.

The online version of this article includes the following source data for figure 5:

**Source data 1.** Replicative lifespan data from the microfluidic imaging used to generate the graphs.

To directly measure intracellular iron levels during aging, we used the fluorescent reporter Phen Green FL (PGFL), which identifies free labile iron ($Fe^{+2}$) (*Gomez-Gallardo et al., 2018*; *Jing et al., 2024*). Control, $P_{GPD1}$-SSD1, *aft1Δ* (a strain that is unable to upregulate cellular iron uptake because it is unable to activate the iron regulon), and CR-treated cells were aged in microfluidic chambers for ~30 hr, generating a population ranging from 3 to ~24 cell divisions in age. Cells were then treated with PGFL, followed by 1,10-phenanthroline (*Gomez-Gallardo et al., 2018*), an iron chelator. The difference in GFP fluorescence before and after PGFL treatment reflects the amount of free intracellular iron. We found that free labile iron accumulates during aging in control cells, albeit with considerable cell-to-cell variability (*Figure 5D and E*). As a control, the *aft1Δ* cells did not accumulate free intracellular iron at any age (*Figure 5D and E*). This result indicates that the accumulation of free intracellular iron during aging is a consequence of activation of the iron regulon. By comparison to the control, the $P_{GPD1}$-SSD1 and CR-treated cells consistently displayed lower intracellular iron concentrations than the control during aging (*Figure 5D and E*). Together, these results indicate that there is an age-dependent iron intake due to activation of the iron regulon, which leads to a higher toxic intracellular free labile iron pool. Meanwhile, age-dependent activation of the iron regulon is reduced during CR or Ssd1 overexpression, limiting the uptake of iron during aging and leading to beneficial effects on longevity.

## Failure to activate the iron regulon during aging is the molecular mechanism whereby Ssd1 overexpression and CR extend RLS

To determine whether reducing activation of the iron regulon is the only mechanism whereby CR and Ssd1 overexpression extend RLS, or whether additional mechanisms are at play, we performed epistasis analysis with the *aft1Δ* strain that is unable to induce the iron regulon. We deleted *AFT1* in

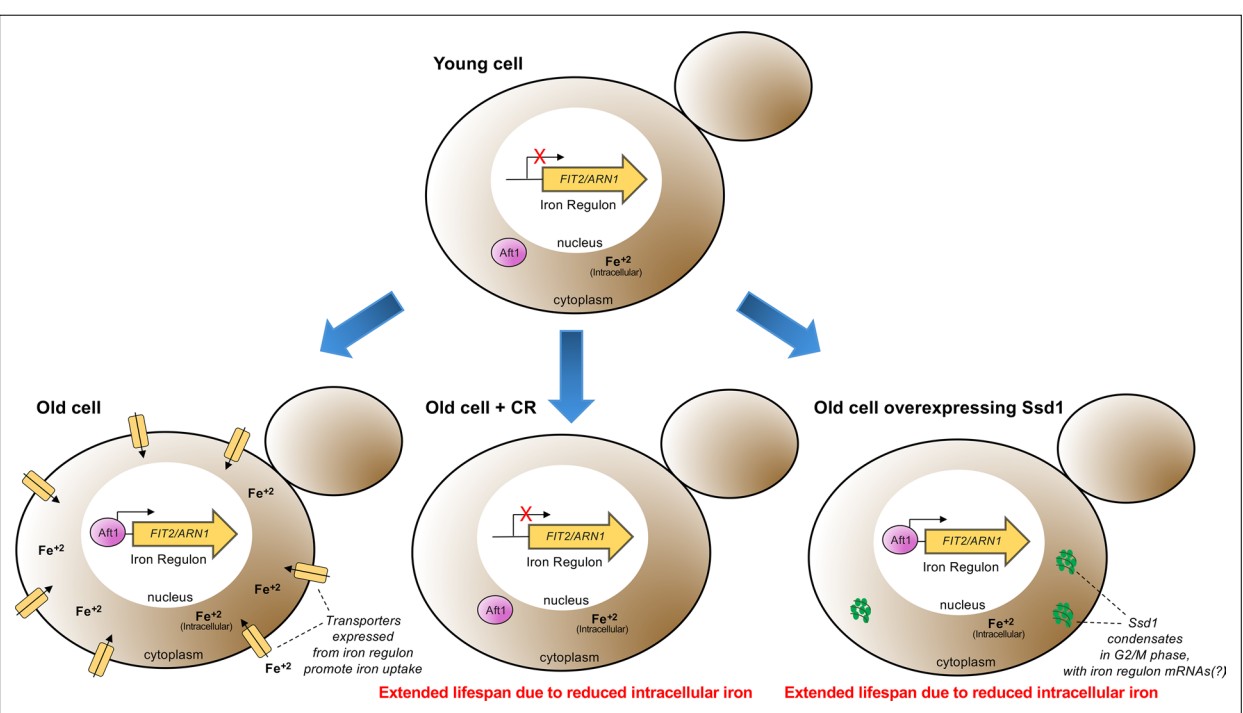

**Figure 6.** Model for molecular mechanism of RLS extension by Ssd1 overexpression and CR. During normal aging, cells have limited intracellular iron, potentially due to mitochondrial dysfunction reducing the assembly of iron-sulfur clusters, which leads to nuclear translocation of Aft1 and activation of the iron regulon. This subsequent deleterious increase in intracellular iron limits lifespan. CR also extends lifespan in a manner dependent on failure to activate the iron regulon; mechanistically, CR indirectly prevents nuclear localization of Aft1, preventing production of the mRNAs from the iron regulon and the subsequent age-dependent increase in deleterious iron accumulation. In aging cells overexpressing Ssd1, transient cytoplasmic Ssd1 condensates appear that correlate with lifespan extension, where the lifespan extension is due to failure to activate the iron regulon preventing deleterious iron accumulation. We speculate that Ssd1 dephosphorylation by Sit4 after $G_1$ phase likely leads to formation of Ssd1 condensates in aging cells overexpressing Ssd1 including mRNAs of the iron regulon genes *FIT2 and ARN1,* while dissolution of Ssd1 condensates occurs prior to cytokinesis upon Cbk1-mediated phosphorylation of Ssd1.

both control and $P_{GPD1}$-SSD1 strains and measured the RLS under both glucose-rich and CR conditions. Given that *aft1Δ* strains have a growth defect independent from its role in iron regulon activation that is reversed by iron supplementation (*Patnaik et al., 2022*), we included 200 μM FeCl$^{+3}$ in our epistasis analyses. We found that *aft1Δ* strains grown in media supplemented with 200 μM FeCl$^{+3}$ were long lived (*Figure 5F*), consistent with our prediction that failure to induce the iron regulon during aging is sufficient to extend lifespan. Importantly, combining *aft1Δ* with $P_{GPD1}$-SSD1 led to the same degree of RLS extension as either *aft1Δ* or $P_{GPD1}$-SSD1 alone (*Figure 5F*). Similarly, growing *aft1Δ* cells under CR led to the same degree of RLS extension as either *aft1Δ* or CR alone (*Figure 5F*). These epistasis analysis results indicate that the absence of activation of the iron regulon during aging, or muted iron regulon activation, is the mechanism whereby CR and Ssd1 overexpression, respectively, extend the RLS (*Figure 6*).

## Discussion

Molecular insights into how anti-aging interventions achieve lifespan extension in any organism are scarce. Here, we report that yeast replicative aging is accompanied by the accumulation of free labile iron, due to activation of the iron regulon – an event that foretells the end of lifespan. We reveal that CR and overexpression of the yeast mRNA binding protein Ssd1 both extend the yeast replicative lifespan by preventing activation of the iron regulon during aging, in turn preventing deleterious age-dependent accumulation of free labile iron that limits lifespan.

### Age-induced cell cycle-regulated Ssd1 condensates and lifespan extension

Lifespan extension by Ssd1 overexpression resulted in the age-induced transient appearance of Ssd1 cytoplasmic foci in a cell cycle stage-dependent manner (*Figures 2 and 3*). The fact that the Ssd1 foci could be dissolved with 1,6-hexanediol and showed fusion properties is consistent with their being condensates. The cell-cycle-dependent nature of formation and dissolution of the age-dependent Ssd1 condensates (*Figure 2A*) is reminiscent of stress-induced formation of Ssd1 foci in cells overexpressing Ssd1 being triggered by Ssd1 dephosphorylation by the Sit4 phosphatase (*Kurischko and Broach, 2017*). Sit4 functions in late $G_1$ phase *Sutton et al., 1991* to promote a timely $G_1$/S transition (*Jiang, 2006*), aligning nicely with the timing of appearance of age-induced Ssd1 foci after $G_1$ phase (*Figure 2A*). Moreover, stress-induced Ssd1 foci in cells overexpressing Ssd1 are dissolved by the Cbk1 kinase (*Kurischko et al., 2011*; *Kurischko and Broach, 2017*), whose peak activity occurs immediately before cytokinesis (*Brace et al., 2011*). This agrees with the fact that virtually all Ssd1 condensates that occurred in aging cells overexpressing Ssd1 dissolved by cytokinesis (*Figure 2A*). It is tempting to speculate that not only is the cell cycle-dependent nature of the appearance of the Ssd1 foci beneficial for longevity, but perhaps also their dissolution by cytokinesis may be important, given that when Ssd1 condensates were retained through mitosis, it coincided with that being the last cell division of their lifespan (*Figure 2A*).

The age-induced Ssd1 foci appear to be a novel type of condensate, because even though we could reproduce the colocalization of overexpressed Ssd1-GFP with stress-induced P-bodies and stress granules (*Kurischko et al., 2011*; *Kurischko and Broach, 2017*), no such colocalization occurred during aging (*Figure 2B–D*). This implies that the lifespan extension that occurs due to Ssd1 overexpression is likely mediated by the Ssd1 condensates themselves rather than by coaggregation with other stress-induced structures. It will be interesting to determine in the future exactly the nature of this novel type of Ssd1 condensate during aging and what kind of stress is triggering its formation during aging but not in young cells (*Figure 2A*). Whatever the mechanism of their formation, age-dependent Ssd1 condensate formation is dependent on the overexpression of Ssd1, as it does not occur in cells expressing Ssd1-GFP from the endogenous promoter during aging, perhaps reflecting a disrupted balance between the molecular ratio of mRNAs and Ssd1 that may promote Ssd1 condensate formation. Notably, previous studies indicate that the fate of Ssd1-bound mRNAs depends on Ssd1's aggregation state, where the mRNAs bound to Ssd1 within a condensate are targeted for degradation or sequestration, while the mRNAs bound to diffusive Ssd1 are targeted for translation (*Kurischko and Broach, 2017*; *Jansen et al., 2009*), consistent with our proposed

mechanism whereby Ssd1 overexpression extends lifespan through beneficial targeted mRNA degradation/segregation to age-induced Ssd1 condensates (*Figure 6*).

## Ssd1 overexpression and CR block the iron regulon to extend lifespan

The induction of Fit2 and Arn1 that occurs during aging (*Figure 4D and E*) was greatly reduced by overexpressing Ssd1 and CR (*Figure 4D and E*), indicating that both anti-aging interventions are reducing activation of the iron regulon during aging. In agreement with induction of the iron regulon during aging, nuclear translocation of Aft1, one of the transcriptional activators of the iron regulon, occurs during aging and is a predictor of on average five remaining cell divisions in the lifespan (*Figure 4B*). Moreover, the earlier the activation of the iron regulon in the lifespan, the shorter the lifespan (*Figure 4C*). Notably, CR prevented nuclear translocation of Aft1 during aging (*Figure 4B*), indicating that cells undergoing CR do not receive the signal to turn on the iron regulon during aging (*Figure 6*). The signal for turning on the iron regulon during aging is likely the depletion of iron-sulfur clusters that occurs during aging as a consequence of mitochondrial dysfunction (*Chen et al., 2020b*; *Veatch et al., 2009*). We speculate that the mechanism whereby CR prevents activation of the iron regulon is related to the fact that CR promotes mitochondrial biogenesis (*López-Lluch et al., 2006*) such that cells undergoing CR may not have depleted levels of iron-sulfur clusters during aging. CR releases the glucose-mediated repression of the heme activator protein (HAP) transcriptional activator complex required for transcription of key nuclear and mitochondrial genes needed for mitochondrial biogenesis (*Lin et al., 2002*), while overexpression of the activating component of the HAP complex, Hap4, itself is sufficient to extend lifespan (*Lin et al., 2002*; *Li et al., 2020*) even in cells grown in glucose. It will be interesting to determine in the future whether overexpression of Hap4 also extends lifespan by preventing the induction of the iron regulon and deleterious accumulation of iron during aging. The ability of CR to activate mitophagy may also play a role in promoting mitochondrial biogenesis (*Mehrabani et al., 2020*).

While Ssd1 overexpression also reduces activation of the iron regulon to extend lifespan, we predict that a different mechanism is at play to block iron regulon activation than during CR. We speculate that the reason why Ssd1 overexpression reduces accumulation of Fit2 and Arn1 during aging (*Figure 4D and E*) is that overexpressed Ssd1 may cause the degradation or sequestration of their mRNAs in the cytoplasmic age-induced Ssd1 condensates (*Figure 6*). Fit2 and Arn1 are not among the known mRNAs that bind to Ssd1 (*Hogan et al., 2008*; *Dutcher and Gasch, 2024*) but this is to be expected as these previous studies were performed in conditions in which the iron regulon, and hence these transcripts, would not have been induced. Also, the Ssd1 mRNA binding partner studies had endogenous levels of Ssd1 not overexpressed which may bind to additional transcripts. Furthermore, it has been reported that Ssd1 can lead to the decay of mRNAs that it does not even bind to *Jansen et al., 2009* and this may be related to the reduced global protein synthesis that occurs due to Ssd1 overexpression (*Hu et al., 2018*). Future studies are required to determine exactly how Ssd1 overexpression leads to reduced levels of Fit2 and Arn1 expression in aged cells.

Once the iron regulon is induced during aging, the production of high-affinity iron transporters such as Fit2 and Arn1 would lead to the subsequent rise in free labile intracellular iron during aging (*Figure 5D and E*). Notably, free labile intracellular iron remains low during aging in strains with Ssd1 overexpression and under CR, consistent with failure to activate the iron regulon (*Figure 5D and E*). Together, these results suggest that age-related increases in iron uptake resulting from activation of the iron regulon are detrimental to replicative lifespan and that CR and Ssd1 overexpression extend lifespan by using different mechanisms to avoid activation of the iron regulon (*Figure 6*). In agreement with their functioning through a shared pathway to extend lifespan, Ssd1 overexpression and CR are epistatic for lifespan extension (*Figure 3*). The fact that pre-activating the iron regulon prevents lifespan extension by either CR or Ssd1 overexpression (*Figure 5A and B*) is consistent with their preventing activation of the iron regulon being their sole mechanism of lifespan extension. Moreover, the key role of the iron regulon in limiting lifespan is apparent from the robust lifespan extension resulting from deletion of *AFT1*, which is also epistatic to the lifespan extension achieved by CR or Ssd1 overexpression (*Figure 5F*).

## Iron levels and lifespan regulation

Activation of the iron regulon during aging has been proposed previously to shorten lifespan by repressing mitochondrial respiration via Cth2, a protein product of the iron regulon targeting the mRNAs of non-essential iron binding proteins for degradation (*Patnaik et al., 2022*). However, we found that deletion of *CTH2* does not extend RLS to the same extent as overexpression of Ssd1 or CR (*Figure 4—figure supplement 4*). Independent studies report that besides deletion of the gene encoding the iron regulon transcription factor *AFT1* (*Yu et al., 2021*), single deletion of the *FIT2* or *ARN1* genes encoding high-affinity iron transporters also extends lifespan (*McCormick et al., 2015*; *Yu et al., 2021*). Although *Saccharomyces cerevisiae* lacks iron export mechanisms, cytosolic iron can be sequestered into storage or iron-sulfur clusters (*Ramos-Alonso et al., 2020*). It is important to note that the iron regulon is composed of several genes with, in some cases, opposite functions; for instance, *CCC1Li et al., 2001* promotes vacuolar storage of iron, while *SMF3* promotes mobilization out of the vacuole (*Portnoy et al., 2000*). Coherently, Ssd1 binds to *SMF3* mRNA (*Dutcher and Gasch, 2024*). As such, overexpressing Ssd1 *may* be modulating iron homeostasis beyond iron intake. Our data reveal that the amount of free labile iron tends to increase as yeast age, whereas the level of free labile iron in cells lacking *ATF1* is low (*Figure 5*), indicating that age-related iron regulon activation drives iron overload. We hypothesize that this excess free labile iron generates oxidative stress, damaging mitochondrial proteins, lipids, and DNA, ultimately impairing respiration. This mechanism is relevant to humans, as age-related iron accumulation in tissues, including the brain, is linked to cognitive decline *Biel et al., 2021* and neurological disorders (*Ndayisaba et al., 2019*; *Cheng et al., 2022*), underscoring its importance for healthspan.

## Effects of CR on human iron metabolism

CR robustly extends lifespan from yeast to mammals (*Fontana et al., 2010*). While it is yet to be demonstrated that CR extends lifespan in humans, it is safe to say that it promotes healthspan (*Caristia et al., 2020*; *Kebbe et al., 2021*). Several studies have shown age-related iron accumulation in human tissues, including the cerebral cortex (*Sato et al., 2022*). Notably, recent studies demonstrate that CR reduces age-related iron accumulation in the brains of rhesus macaques (*Kastman et al., 2010*) and rodents (*Xu et al., 2008*; *Cook and Yu, 1998*). In humans, CR and intermittent fasting improve memory and cognitive function during aging (*Witte et al., 2009*; *Leclerc et al., 2020*). Whether these benefits stem from reduced iron accumulation remains to be investigated. We asked if there is any evidence that CR ameliorates toxic iron accumulation in humans. To investigate this, we analyzed published data. While we were unable to find proteomic data from specific tissues such as human brain under CR, we assessed the blood/plasma proteomics change of two studies from overweight donors subjected to CR leading to weight loss and stabilization (*Goudswaard et al., 2023*; *Capel et al., 2009*), two studies identifying proteins that correlate with age (*Siino et al., 2022*; *Lehallier et al., 2019*) and one study analyzing protein signatures of centenarians (*Sebastiani et al., 2021*). Transferrin (TF), a blood plasma glycoprotein that binds and transports iron throughout the bloodstream to cells and tissues in the body, was upregulated in CR (*Goudswaard et al., 2023*; *Capel et al., 2009*), while decreases with age (*Siino et al., 2022*; *Lehallier et al., 2019*), suggesting that an increase in TF may reflect enhanced capacity to bind and sequester free iron, reducing its toxic potential. CR also upregulates hepcidin (HAMP) (*Goudswaard et al., 2023*), a peptide hormone that serves as the master regulator of systemic iron homeostasis by inhibiting iron absorption and release from cells, consistent with people under CR potentially absorbing less iron from their diet (*Pagani et al., 2019*). Indeed, it has been shown that low levels of hepcidin lead to chronic liver disease as a result of iron accumulation (*Milic et al., 2016*). HAMP was slightly but not significantly downregulated with age in both studies (*Siino et al., 2022*; *Lehallier et al., 2019*). On the other hand, the transferrin receptor (TFRC) was downregulated in people under CR (*Goudswaard et al., 2023*; *Capel et al., 2009*). TFRC is present on cell surfaces to mediate iron uptake through transferrin-iron complexes *Gammella et al., 2021*. A decrease under CR suggests that human cells reduce iron uptake from the plasma, limiting excess iron storage. TFRC is downregulated in centenarians *Sebastiani et al., 2021*, and is slightly, but not significantly, increased with age (*Siino et al., 2022*). Downregulation of ferritin heavy chain (FTH1) occurs in CR (*Goudswaard et al., 2023*; *Capel et al., 2009*), given that FTH1 is a component of ferritin, the iron storage protein, this downregulation could be a consequence of putative lower iron abundance in the plasma of people under CR. These findings highlight the potential for targeting

iron-related proteins to influence iron homeostasis and lifespan in humans. However, further studies will be needed to determine whether modulating the expression of these genes or proteins in humans could help reduce age-related iron accumulation in tissues, offering potential therapeutic strategies for conditions associated with iron dysregulation.

## Methods

**Key resources table**

| Reagent type (species) or resource | Designation | Source or reference | Identifiers | Additional information |
|---|---|---|---|---|
| Strain, strain background (*Saccharomyces cerevisiae*) | BY4741 background, parental ssd1::NATMX | *Kurischko et al., 2011* | FLY2184 | Parental strain for all others in study |
| Strain, strain background (*Saccharomyces cerevisiae*) | SSD1-GFP (GPD1 promoter) | This paper; pAG415-GPD1-SSD1-GFP (FLE1019 in *Kurischko et al., 2011*) | See *Supplementary file 1* | Constructed by homologous recombination |
| Gene (*Saccharomyces cerevisiae*) | SSD1 | SGD | SGD:S000002701 | Systematic Name: YDR293C |
| Sequence-based reagent | GPD1 promoter | SGD | SGD:S000003709 (GPD1 gene) | Used as constitutive promoter |
| Strain, strain background (*Saccharomyces cerevisiae*) | PGPD1-GFP-AFT1 | This paper; pRS plasmid *Baker Brachmann et al., 1998* | See *Supplementary file 1* | Homologous recombination |
| Strain, strain background (*Saccharomyces cerevisiae*) | Gene deletions, mCherry and GFP tagging strains | *Longtine et al., 1998* | See *Supplementary file 1* | Homologous recombination |
| Chemical compound, drug | Yeast Nitrogen Base (no amino acids, no ammonium sulfate) | Difco / BD Biosciences | 233520 | Component of SCD medium |
| Chemical compound, drug | Complete drop-out mix without YNB | US Biological | D9515 | Component of SCD medium |
| Chemical compound, drug | Ammonium sulfate | Fisher Chemical | A702-3 | Component of SCD medium |
| Chemical compound, drug | D-glucose | Sigma-Aldrich | G8270 | Component of SCD and YPD media |
| Chemical compound, drug | Dextrose (for YPD) | Grainger | 31GE61, 31GC58 | YPD component |
| Chemical compound, drug | Artemisinin | Thermo Scientific | J65406.03 | Used in supplementation experiments |
| Other | Automated dissection/screening chip | iBio chips | Not specified | Used for RLS analysis |
| Chemical compound, drug | Iron(III) chloride | Sigma-Aldrich | 157740–100 G | Iron supplementation |
| Chemical compound, drug | Bathophenanthrolinedisulfonic acid disodium salt hydrate (BPS) | Sigma-Aldrich | 146617–1 G | Iron chelator in media |
| Other | Filter bottle, 0.22 um PES, 54.5 cm² | Corning | 431098 | Media sterilization |
| Other | EVOS II microscope | Thermo Fisher | Not specified | Used for all imaging |
| Other | EVOS Light Cube GFP 2.0 | Invitrogen Thermo Fisher | AMEP4951 | FITC/GFP filter, fluorescence imaging |
| Other | EVOS Light Cube TxRed 2.0 | Invitrogen Thermo Fisher | AMEP4955 | TxRed/mCherry filter, fluorescence imaging |
| Chemical compound, drug | Ascorbic acid | LabChem | LC115309 | 20 mM in fluorescence media to reduce photobleaching |
| Software, algorithm | ImageJ | NIH | https://imagej.nih.gov | Manual image analysis |
| Software, algorithm | GraphPad Prism | GraphPad | https://graphpad.com | Plotting and statistics |

*Continued on next page*

*Continued*

| Reagent type (species) or resource | Designation | Source or reference | Identifiers | Additional information |
|---|---|---|---|---|
| Software, algorithm | Excel | Microsoft | https://office.com | Data management & plotting |
| Software, algorithm | OASIS 2 | *Han et al., 2016* | https://sbi.postech.ac.kr/oasis2/ | RLS survival curves |
| Chemical compound, drug | Phen Green FL, Diacetate | Thermo Fisher | P6763 | Iron measurement, 5 Âμg/mL |
| Chemical compound, drug | 1,10-Phenanthroline | Sigma-Aldrich | 131377–5 G | Iron chelation, 1 mM |
| Other | LSR II | BD Biosciences | Not specified | For FITC/TxRed flow cytometry |

## Yeast strains

All strains were derived from BY4741 (*Baker Brachmann et al., 1998*). *SSD1* was replaced by a NatMX cassette (FLY2184, gift from Cornelia Kurischko) (*Kurischko et al., 2011*). To generate *SSD1*-eGFP fusions, we integrated PCR products amplified from plasmid FLE1019 (*Kurischko et al., 2011*) (pAG415-$P_{GPD1}$-*SSD1*) and pFA6a-GFP(S65T)-KanMX6 (*Longtine et al., 1998*) via homologous recombination. These constructs included the *GPD1* or *SSD1* promoter, respectively, and were integrated into the genome at the *SSD1* locus by homologous recombination.

Gene deletions and the addition of the remaining fluorescent tags (except for $P_{GPD1}$-GFP-*AFT1*) were generated by homologous recombination with PCR products from plasmids developed by Pringle et al. (*Longtine et al., 1998*). The $P_{GPD1}$-GFP-*AFT1* strain was constructed by homologous recombination of a PCR product from a pRS plasmid containing $P_{GPD1}$-eGFP (*Chee and Haase, 2012*). All strains used in this study are cataloged in *Supplementary file 1*.

## Replicative lifespan analyses

Replicative lifespan (RLS) analyses were performed as follows. Unless otherwise indicated, cells were grown overnight in Synthetic Complete Media (SCD) composed of yeast nitrogen base without amino acids and ammonium sulfate (Difco 233520), complete drop-out mix without YNB (US Biological D9515), ammonium sulfate (Fisher Chemical A702-3), and glucose (Sigma-Aldrich G8270). Cultures were grown to an optical density (OD)$_{600}$ between 0.2 and 0.6, then diluted to an OD$_{600}$ of 0.1, vortexed for 30 s, and loaded into microfluidic (described below in more detail) and as we recently described (*Gutierrez and Tyler, 2024*). For experiments involving artemisinin (Art) supplementation, cells were cultured in yeast peptone dextrose (YPD) media containing the indicated concentration of dextrose (Grainger 31GE61; 31GC58), with artemisinin (Thermo Scientific J65406.03) added directly to fresh YPD prior to sterilization. RLS was determined using microfluidic chips from iBiochips according to the manufacturer's instructions. Where indicated, iron(III) chloride (Sigma 157740–100 G) and bathophenanthrolinedisulfonic acid disodium salt hydrate (BPS; Sigma 146617–1 G) were added directly to SCD media before sterilization using a Corning filter bottle system with a 0.22 μm pore 54.5 cm² PES membrane (Corning 431098), at the specified final concentrations. All experimental conditions and compound concentrations are detailed in the relevant figure legends.

Each strain was analyzed in the microfluidic chamber under non-fluorescent conditions in three independent experiments, each yielding consistent results, with 30–50 cells counted per condition or strain in each experiment. Cells that were lost from the microfluidic traps during the course of the experiment were recorded as censored. For strains expressing fluorescent markers, experiments were conducted under the appropriate fluorescent illumination, also in three independent replicates with consistent results, with 20–40 cells counted per experiment. For these fluorescent strains, the total number of cells from all three experiments is presented in the histograms, and only cells that completed their full replicative lifespan were included in the analysis, with cells lost from the traps excluded. Replicative lifespan survival curves were generated using the OASIS 2 platform (https://sbi.

postech.ac.kr/oasis2/) as described by *Han et al., 2016*, and p-values for RLS curves were calculated using the log-rank test.

## Flow cytometry

The settings used on an LSR II flow cytometer were for detection of FITC (excitation 494, emission 520) and TxRed (excitation at 595 nm, emission at 613 nm). For each sample, data from at least 10,000 events were collected, and fluorescence intensity was quantified using pulse area (integral) measurements.

## Image collection

All imaging in this study, including replicative lifespan (RLS) experiments, was performed using an EVOS II microscope (Thermo Fisher). For colocalization experiments, Ssd1-GFP was imaged using the EVOS Light Cube GFP 2.0 (Invitrogen Thermo Fisher, AMEP4951) with light intensity set to 0.053, exposure time to 0.047 s, and gain to 1. Hsp104-mCherry, Edc3-mCherry, and Pab1-mCherry were imaged using the EVOS Light Cube TxRed 2.0 (Invitrogen Thermo Fisher, AMEP4955) with light intensity at 0.6, exposure time at 0.06 s, and gain at 5. For RLS experiments without fluorescence microscopy, imaging was performed at ×40 magnification on V-shaped chips (iBiochips; *Gutierrez and Tyler, 2024*), with images acquired every 20 min for 96 hr. Brightfield illumination was adjusted individually for each experiment, and z-stacks were collected over a physical length of 19 μm in 5 steps of 3.8 μm each. When RLS was measured in conjunction with fluorescence microscopy, imaging was conducted at ×60 magnification with immersion oil on Y-shaped AD chips (iBiochips; *Gutierrez and Tyler, 2024*), with images taken every 20 min for 96 hr. Ssd1-GFP was imaged as above, but with an exposure time of 0.045 s. Hsp104-mCherry, Edc3-mCherry, and Pab1-mCherry were imaged with light intensity at 0.3, exposure time at 0.06 s, and gain at 5. Fit2-mCherry and Arn1-mCherry were imaged with light intensity at 0.105, exposure time at 0.05 s, and gain at 4. Aft1-GFP was imaged with light intensity at 0.053, exposure time at 0.04 s, and gain at 1. For measurements involving the GFP channel in PGFL experiments, the same light cube was used with light intensity at 0.053, exposure time at 0.07 s, and gain at 1. In general, higher exposure times and lower light intensities were preferred to minimize phototoxicity, and all media used for fluorescence imaging was supplemented with 20 mM ascorbic acid (LabChem LC115309) to reduce photobleaching. Fluorescence imaging experiments used z-stacks spanning 8.5 μm in 5 steps of 1.7 μm. All imaging parameters were optimized to achieve the desired temporal resolution while minimizing phototoxicity.

## Image analysis and data representations

The number of cell divisions in each RLS was determined manually by analyzing images captured at ×40 magnification every 20 min, using the image sequence function in ImageJ. Fluorescence microscopy images were also analyzed manually in ImageJ, with appropriate background corrections applied. For Ssd1 foci quantification, a minimum threshold of 200-pixel intensity above background was used to score foci; structures below this threshold were disregarded. Multiple foci per cell cycle were recorded if two or more foci were observed simultaneously in the same cell, or if a focus disappeared and a new focus appeared in subsequent images within the same cell cycle. For single-cell trajectory histograms, individual values for fluorescence intensity or foci number, depending on the experiment, were compiled from three independent experimental repeats and visualized in Excel by replacing values with the corresponding color codes in the cells. Dot plots were generated using Prism, while box-and-whisker plots were created in Excel. Statistical significance for dot plots and box-and-whisker plots was assessed using the two-sided Student's t-test. All analyses were performed in at least three independent experiments with reproducible results, and the combined data are presented in the figures.

## Iron measurements

Intracellular free iron ($Fe^{+2}$) levels were measured using Phen Green FL, Diacetate, cell permeant (Thermo Fisher P6763), following a yeast-adapted protocol as described by *Gomez-Gallardo et al., 2018*. Briefly, cells were aged in the microfluidic chamber as previously described, and at approximately 30 hr, the chip was loaded with 5 μg/mL Phen Green FL in DMSO. GFP fluorescence was measured using the EVOS Light Cube GFP 2.0 (Invitrogen Thermo Fisher, AMEP4951). The chip was then loaded with 1 mM of the iron chelator 1,10-phenanthroline (Sigma 131377–5 G), and fluorescence

was measured again. The ratio of fluorescence before and after chelator addition is proportional to the concentration of intracellular free labile iron in the cell.

## Acknowledgements

We are indebted to Cornelia Kurischko for extensive thoughtful discussions about Ssd1 condensates, and the Weill Cornell Flow Cytometry Core for technical support. JKT is supported by NIH grants R01 AG079883 and R35 GM139816.

## Additional information

### Funding

| Funder | Grant reference number | Author |
|---|---|---|
| NIH Office of the Director | RO1 AG079883 | Jessica K Tyler |
| NIH Office of the Director | GM139816 | Jessica K Tyler |

The funders had no role in study design, data collection and interpretation, or the decision to submit the work for publication.

### Author contributions

J Ignacio Gutierrez, Conceptualization, Data curation, Formal analysis, Investigation, Methodology, Writing – original draft; Claudia Edgar, Formal analysis; Jessica K Tyler, Conceptualization, Funding acquisition, Writing – review and editing

### Author ORCIDs

J Ignacio Gutierrez https://orcid.org/0000-0002-9017-8384
Jessica K Tyler https://orcid.org/0000-0001-9765-1659

Reviewer #1 (Public review): https://doi.org/10.7554/eLife.108892.3.sa1
Reviewer #2 (Public review): https://doi.org/10.7554/eLife.108892.3.sa2
Reviewer #3 (Public review): https://doi.org/10.7554/eLife.108892.3.sa3
Author response https://doi.org/10.7554/eLife.108892.3.sa4

## Additional files

### Supplementary files

Supplementary file 1. Table of Yeast strains used in this study.

MDAR checklist

### Data availability

Figure 1- source data 1, Figure 2- source data 1, Figure 3- source data 1, Figure 4- source data 1, and Figure 5- source data 1 contain the numerical data used to generate the figures.

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
