## [Editor Report · eLife Assessment]

This **important** study uses innovative microfluidics-based single-cell imaging to monitor replicative lifespan, protein localization, and intracellular iron levels in aging yeast cells. The evidence for the proposed role of Ssd1 and reduced nutrients for lifespan through limiting iron uptake is **convincing**, even though some mechanistic details remain unclear. This work will be of interest to cell biologists working on aging and iron metabolism.

---

## [Referee Report · Reviewer #1 (Public review)]

Summary:

Overexpression of the mRNA binding protein Ssd1 was shown before to expand the replicative lifespan of yeast cells, whereas ssd1 deletion had the opposite effect. Here, the authors provide initial evidence that overproduced Ssd1 might act via sequestration of mRNAs of the Aft1/2-dependent iron regulon. Ssd1 overexpression restricts activation of the iron regulon and limits accumulation of Fe2+ inside cells, thereby likely lowering oxidative damage. The effects of Ssd1 overexpression and calorie restriction on lifespan are epistatic, suggesting that they might act through the same pathway.

Strengths:

The study is well-designed and involves analysis of single yeast cells during replicative aging. The findings are well displayed and largely support the derived model, which also has implications on lifespan of other organisms including humans.

Weaknesses:

The model is largely supported by the findings, however they remain correlative at the same time. Whether the knockout of ssd1 shortens lifespan by increased intracellular Fe2+ levels is unknown and the shortened lifespan might be caused by different Ssd1 functions. The finding that increased Ssd1 levels form condensates in a cell-cycle dependent is interesting, yet the role of the condensates in lifespan expansion remains untested and unlinked.

Comments on revisions:

In their revised version and response letter the authors have largely addressed my previous concerns. I would have liked to see an experimental response to some of the points of criticism, but I accept that they have been addressed purely in writing. There are some aspects that should be further elaborated by the authors. I agree that determining the mRNAs that co-sequester with Ssd1 foci will be part of an independent study, yet whether Ssd1 foci are relevant for lifespan expansion remains unclear and I would have hoped for some more detailed consideration on this point in the discussion section. Similarly, it should be clearly stated that the impact of Ssd1 overexpression is unlinked from the cellular function of Ssd1 produced at authentic levels and that the short-lived phenotype of a ssd1 knockout is likely not caused by overactivation of the iron regulon (based on the author´s reply). I will appreciate it if the authors include these aspects more clearly in the discussion.

---

## [Referee Report · Reviewer #2 (Public review)]

This manuscript describes the use of a powerful technique called microfluidics to elucidate the mechanisms explaining how overexpression (OE) of Ssd1 and caloric restriction (CR) in yeast extend replicative lifespan (RLS). Microfluidics measures RLS by trapping cells in chambers mounted to a slide. The chambers hold the mother cell but allow daughters to escape. The slide, with many chambers, is recorded during the entire process, roughly 72 hours, with the video monitored afterwards to count how many daughters each of the trapped mothers produces. The power of the method is what can be done with it. For example, the entire process can be viewed by fluorescence so that GFP and mCherry-tagged proteins can be followed as cells age. The budding yeast is the only model where bona fide replicative aging can be measured, and microfluidics is the only system that allows protein localization and levels to be measured in a single cell while aging. The authors do a wonderful job of showing what this combination of tools can do.

The authors had previously shown that Ssd1, an mRNA-binding protein, extends RLS when overexpressed. This was attributed to Ssd1 sequestering away specific mRNAs under stress, likely leading to reduced ribosomal function. It remained completely unknown how Ssd1 OE extended RLS. The authors observed that overexpressed, but not normally expressed, Ssd1 formed cytoplasmic condensates during mitosis that are resolved by cytokinesis. When the condensates fail to be resolved at the end of mitosis, this signals death.

It has become clear in the literature that iron accumulation increases with age within the cell. The transcriptional programs that activate the iron regulon also become elevated in aging cells. This is thought to be due to impaired mitochondrial function in aging cells, with increased iron accumulation as an attempt at restoring mitochondrial activity. The authors show that Ssd1 OE and CR both reduce the expression of the iron regulon. The data presented indicate that iron accumulation shortens RLS: deletion of iron regulon components extends RLS, and adding iron to WT cells decreases RLS, but not when Ssd1 is overexpressed or when cells are calorically restricted. Interestingly, iron chelation using BPS has no impact on WT RLS, but decreases the elevated RLS in CR cells and cells overexpressing Ssd1. It was not initially clear why iron chelation would inhibit the extended lifespan seen with CR and Ssd1 OE. This was addressed by an experiment where it was shown that the iron regulon is induced (FIT2 induction) when iron is chelated. Thus, the detrimental effects of induction of the iron regulon by BPS and iron accumulation on RLS cannot be tempered by Ssd1 OE and CR once turned on.

Comments on Revised Version:

I am content with the authors' responses to my prior comments.

---

## [Referee Report · Reviewer #3 (Public review)]

In this paper, the authors investigate how the RNA-binding protein Ssd1 and calorie restriction (CR) influence yeast replicative lifespan, with a particular focus on age-dependent iron uptake and activation of the iron regulon. For this, they use microfluidics-based single-cell imaging to monitor replicative lifespan, protein localization, and intracellular iron levels across aging cells. They show that both Ssd1 overexpression and CR act through a shared pathway to prevent the nuclear translocation of the iron-regulon regulator Aft1 and the subsequent induction of high-affinity iron transporters. As a result, these interventions block the age-related accumulation of intracellular free iron, which otherwise shortens lifespan. Genetic and chemical epistasis experiments further demonstrate that suppression of iron regulon activation is the key mechanism by which Ssd1 and CR promote replicative longevity.

Overall, the paper is technically rigorous, and the main conclusions are supported by a substantial body of experimental data. The microfluidics-based assays in particular provide compelling single-cell evidence for the dynamics of Ssd1 condensates and iron homeostasis.

My main concern, however, is that the central reasoning of the paper-that Ssd1 overexpression and CR prevent the activation of the iron regulon-appears to be contradicted by previous findings, and the authors may actually be misrepresenting these studies, unless I am mistaken. In the manuscript, the authors state on two occasions:

"Intriguingly, transcripts that had altered abundance in CR vs control media and in SSD1 vs ssd1∆ yeast included the FIT1, FIT2, FIT3, and ARN1 genes of the iron regulon (8)"

"Ssd1 and CR both reduce the levels of mRNAs of genes within the iron regulon: FIT1, FIT2, FIT3 and ARN1 (8)"

However, reference (8) by Kaeberlein et al. actually says the opposite:

"Using RNA derived from three independent experiments, a total of 97 genes were observed to undergo a change in expression >1.5-fold in SSD1-V cells relative to ssd1-d cells (supplemental Table 1 at http://www.genetics.org/supplemental/). Of these 97 genes, only 6 underwent similar transcriptional changes in calorically restricted cells (Table 2). This is only slightly greater than the number of genes expected to overlap between the SSD1-V and CR datasets by chance and is in contrast to the highly significant overlap in transcriptional changes observed between CR and HAP4 overexpression (Lin et al. 2002) or between CR and high external osmolarity (Kaeberlein et al. 2002). Intriguingly, of the 6 genes that show similar transcriptional changes in calorically restricted cells and SSD1-V cells, 4 are involved in iron-siderochrome transport: FIT1, FIT2, FIT3, and ARN1 (supplemental Table 1 at http://www.genetics.org/supplemental/)."

Although the phrasing might be ambiguous at first reading, this interpretation is confirmed upon reviewing Matt Kaeberlein's PhD thesis: https://dspace.mit.edu/handle/1721.1/8318

(page 264 and so on)

Moreover, consistent with this, activation of the iron regulon during calorie restriction (or the diauxic shift) has also been observed in two other articles:

https://doi.org/10.1016/S1016-8478(23)13999-9

https://doi.org/10.1074/jbc.M307447200

Taken together, these contradictory data might blur the proposed model and make it unclear how to reconcile the results.

Comments on revisions:

The authors successfully addressed my requests and concerns

---

## [Author Response]

The following is the authors’ response to the original reviews.

**Reviewer #2 (Public review):**
(1) Why would BPS not reduce RLS in WT cells? The authors could test whether OE of FIT2 reduces RLS in WT cells.

Our data indicate that the iron regulon gets turned on naturally in old cells, presumably due to reduced iron sensing, limiting their lifespan. Although we haven’t tested it experimentally, BPS would also turn on the iron regulon presumably in wild type cells and therefore would have a redundant effect with the activation of the iron regulon that occurs naturally during normal aging. It may be interesting in the future to see if higher levels of BPS can shorten the lifespan of wildtype cells. Similarly, we would predict that overexpression of FIT2 may reduce the lifespan, as its deletion has been shown to extend RLS.

(2) The authors should add a brief explanation for why the GDP1 promoter was chosen for Ssd1 OE.

We used the same promoter that was used to overexpress Ssd1 in all previous studies. This is now stated in the text along with the relevant citations.

(3) On page 12, growth to saturation was described as glucose starvation. This is more accurately described as nutrient deprivation. Referring to it as glucose starvation is akin to CR, which growing to saturation is not. Ssd1 OE formed condensates upon saturation but not in CR. Why do the authors think Ssd1 OE did not form condensates upon CR?

Too mild a stress?

This is a fair comment, and we have now changed glucose starvation to nutrient deprivation, as it is more accurate. The effects of nutrient starvation are profound: the cell cycle stops, autophagy is induced, cells undergo the diauxic shift, metabolism changes. None of these changes occur during calorie restriction (0.05% glucose) such that it is not too surprising that Ssd1 does not form condensates during CR. We speculate that the stress is just too mild.

(4) The authors conclude that the main mechanism for RLS extension in CR and Ssd1 OE is the inhibition of the iron regulon in aging cells. The data certainly supports this. However, this may be an overstatement as other mutations block CR, such as mutations that impair respiration. The authors do note that induction of the iron regulon in aging cells could be a response to impaired mitochondrial function. Thus, it seems that the main goal of CR and Ssd1 OE may be to restore mitochondrial function in aging cells, one way being inactivation of the iron regulon. A discussion of how other mutations impact CR would be of benefit.

While some labs have shown that respiration impacts CR, this is not the case in other studies. For example, an impactful paper by Kaeberlein et al., PLOS Genetics 2005 showed that CR does extend lifespan in respiratory deficient strains using many different strain backgrounds.

(5) The cell cycle regulation of Ssd1 OE condensates is very interesting. There does not appear to be literature linking Ssd1 with proteasome-dependent protein turnover. Many proteins involved in cell cycle regulation and genome stability are regulated through ubiquitination. It is not necessary to do anything here about it, but it would be interesting to address how Ssd1 condensates may be regulated with such precision.

we see no evidence of changes in Ssd1 protein intensity during the cell cycle. The difference therefore we speculate is at the post translational level rather than Ssd1 degradation and there are known cell cycle regulated phosphatase and kinase that regulates Ssd1 phosphorylation and condensation state whose timing of function match when the Ssd1 condensates appear and dissolve in the cell cycle. We have now discussed this and elude to it in the model.

(6) While reading the draft, I kept asking myself what the relevance to human biology was. I was very impressed with the extensive literature review at the end of the discussion, going over how well conserved this strategy is in yeast with humans. I suggest referring to this earlier, perhaps even in the abstract. This would nail down how relevant this model is for understanding human longevity regulation.

Thank you, we have now mentioned in the abstract the relevance to human work.

In conclusion, I enjoyed reading this manuscript, describing how Ssd1 OE and CR lead to RLS increases, using different mechanisms. However, since the 2 strategies appear to be using redundant mechanisms, I was surprised that synergism was not observed.

We thank the reviewer for their kind comment. We propose that Ssd1 overexpression impacts the levels of the iron regulon transcripts, which would be downstream of the point in the pathway that is affected by CR, i.e., nuclear localization of Aft1. The lack of synergy fits with this model, as Ssd1 overexpression cannot impact the iron regulon transcripts if they are not induced due to CR. We have now improved the model to make the impact of these different anti-aging interventions on activation of the iron regulon more clear.

**Reviewer #3 (Public review):**
My main concern is that the central reasoning of the paper-that Ssd1 overexpression and CR prevent the activation of the iron regulon-appears to be contradicted by previous findings, and the authors may actually be misrepresenting these studies, unless I am mistaken. In the manuscript, the authors state on two occasions:"Intriguingly, transcripts that had altered abundance in CR vs control media and in SSD1 vs ssd1∆ yeast included the FIT1, FIT2, FIT3, and ARN1 genes of the iron regulon (8)""Ssd1 and CR both reduce the levels of mRNAs of genes within the iron regulon: FIT1, FIT2, FIT3 and ARN1 (8)"However, reference (8) by Kaeberlein et al. actually says the opposite:"Using RNA derived from three independent experiments, a total of 97 genes were observed to undergo a change in expression >1.5-fold in SSD1-V cells relative to ssd1d cells (supplemental Table 1 at http://www.genetics.org/supplemental/). Of these 97 genes, only 6 underwent similar transcriptional changes in calorically restricted cells (Table 2). This is only slightly greater than the number of genes expected to overlap between the SSD1-V and CR datasets by chance and is in contrast to the highly significant overlap in transcriptional changes observed between CR and HAP4 overexpression (Lin et al. 2002) or between CR and high external osmolarity (Kaeberlein et al. 2002). Intriguingly, of the 6 genes that show similar transcriptional changes in calorically restricted cells and SSD1-V cells, 4 are involved in ironsiderochrome transport: FIT1, FIT2, FIT3, and ARN1 (supplemental Table 1 at http://www.genetics.org/supplemental/)."Although the phrasing might be ambiguous at first reading, this interpretation is confirmed upon reviewing Matt Kaeberlein's PhD thesis: https://dspace.mit.edu/handle/1721.1/8318 (page 264 and so on).Moreover, consistent with this, activation of the iron regulon during calorie restriction (or the diauxic shift) has also been observed in two other articles:
https://doi.org/10.1016/S1016-8478(23)13999-9

https://doi.org/10.1074/jbc.M307447200
Taken together, these contradictory data might blur the proposed model and make it unclear how to reconcile the results.

We thank the reviewer for pointing this out. Upon further consideration, we have now removed all mention of this paper from our manuscript as it is irrelevant to our situation, because the mRNA abundance studies during CR or with and without Ssd1 were not performed in situations in which the iron regulon is even activated such as aging, so there would not be any opportunity to detect reduced transcript levels due to CR or Ssd1 presence. Also, none of these studies were performed with Ssd1 overexpression which is the situation we are examining. Our data clearly show that Ssd1 overexpression and CR reduced / prevented, respectively, production of proteins from the iron regulon during aging.

We do not feel that the iron regulon being activated by nutrient depletion at the diauxic shift is a fair comparison to the situation in cells happily dividing during CR. The levels of nutrient deprivation used in those studies have profound effects including arresting cell growth, activating autophagy, altering metabolism. The levels of CR that we use (0.05% glucose) does not activate any of these changes nor the iron regulon in young cells or old cells (Fig. 4).

**Reviewer #1 (Recommendations for the authors):**
(1) The role of Ssd1 condensate formation in mRNA sequestration and lifespan expansion remains unclear. Thus, the study involves two parts (Ssd1 condensate formation and lifespan expansion via limiting Fe2+ accumulation), which are poorly linked. The study will therefore benefit from further data linking the two aspects.

Future experiments are planned to determine what mRNAs reside in the age-induced Ssd1 overexpression condensates, to determine if they include the iron regulon transcripts. This will require us to optimize isolation of old cells and isolation of the Ssd1 condensates from them, and is beyond the scope of the present study.

(2) The beneficial effects of Ssd1 overexpression and calorie restriction (CR) on lifespan are epistatic, yet the claim that both experimental conditions act via the same pathway should be further documented. It is recommended to combine Ssd1 overexpression with a well-defined condition that expands lifespan through a mechanism not involving changes in Fe2+ levels. A further increase in lifespan upon combining such conditions would at least indirectly support the authors' claim.

We have more than epistatic evidence to indicate that Ssd1 overexpression and CR are in the same pathway. Ssd1 overexpression and CR result in failure to properly induce the iron regulon during aging and subsequent reduced levels of iron, resulting in lifespan extension, supporting that they act via the same pathway. We do appreciate the point though and epistasis analyses are on our list for future studies.

(3) It is highly recommended to analyze ssd1 knockout cells: Is the shortened lifespan caused by intracellular Fe2+ accumulation, as predicted by the model? Does the knockout lead to an overactivation of the iron regulon? Such analysis will also document the physiological relevance of authentic Ssd1 levels in controlling yeast lifespan. The authors could test this possibility by determining intracellular Fe2+ levels (as done in Figure 5) and testing whether the mutant cells are partially rescued by the presence of an iron chelator (as done in Figure 5C).

We don’t think the normal role of Ssd1 is to sequester the iron regulon mRNAs to prevent its activation, given that wild type yeast with endogenous Ssd1 activates the iron regulon during aging. Rather, the failure to activate the iron regulon during aging is unique to when Ssd1 is overexpressed not at endogenous Ssd1 levels. As such, it may not be the case that the short lifespan of ssd1 yeast is due to iron accumulation (if that happens); yeast lacking SSD1 also have cell wall biogenesis problems and the defects in cell wall biogenesis shorten the replicative lifespan (Molon et al., Biogerentology 2018 PMID 29189912).

(4) Figure 4: The authors could not analyze the impact of Ssd1 overexpression on the localization of GFP-Aft1 due to synthetic sickness. This was not observed under calorie restriction (CR) conditions and is therefore unexpected. Why should Ssd1 overexpression and CR have such diverse impacts on cellular physiology when combined with GFP-Aft1? Isn`t that observation arguing against CR and increased Ssd1 levels acting through the same pathway? A further clarification of this point is necessary.

Without further experimentation, we can only speculate that cellular changes that are unique to overexpression of Ssd1 and not shared with CR cause a negative interaction with GFP-Aft1. Of note, Aft1 has functions in addition to its role in activating the iron regulon (aft1∆ strains have a growth defect independent from its role in iron regulon activation [27]) and we have shown previously that overexpressed Ssd1 has a reduction in global protein translation. Future experiments would be necessary to delineate the basis for this synthetic sickness.

(5) Lowering Fe2+ levels upon Ssd1 overexpression is predicted to reduce oxidative stress. It is suggested to determine ROS levels upon Ssd1 overexpression to bolster that point.

This is a great suggestion. The lowering of Fe2+ in the Ssd1 mutants is something that happens at the end of the lifespan and therefore we would need to do experiments to detect reduced ROS using a live dye on our microfluidics platform. We are not aware of any live fluorescent reporters of ROS.

**Reviewer #2 (Recommendations for the authors):**
(1) Page 6, 7th line of Replicative lifespan analyses, there is a double bracket.

This has been corrected. Thank you

(2) Page 18, line 6 of "failure to activate..." section, "revered" should be replaced with "reversed".

This has been corrected. Thank you

(3) Page 23, fix writing on line 2 of "Effects of CR..." section.

This has been corrected. Thank you

(4) Page 24, Author contributions section, replace "performed devised" with "designed".

This has been corrected. Thank you

**Reviewer #3 (Recommendations for the authors):**
(1) Figure 3C: The panel legend is somewhat confusing due to the color scheme and the scattering of labels across panels. A more consistent labeling strategy would help readability.

We agree, and the labelling has now been improved. Thank you.

(2) Figure 3D vs Figure 3B: it appears that Fit2 activation occurs substantially earlier than Aft1 translocation, which reduces the predictive value of Fit2 compared to Aft1. This is puzzling given that Fit2 is expected to be a direct target of Aft1. Could this discrepancy be related to the thresholding used for Fit2-mCherry display? The color scale in Figure 3D is also somewhat misleading, as most of the segments appear greenish. A continuous color gradient, perhaps restricted to the [10-120] interval, might give a clearer picture of iron regulon activation.

For the Aft1-mcherry experiment, we are only able to accurately annotate nuclear localization when Aft1 has been fully (or mostly) translocated into the nucleus from the cytoplasm such that this data is likely to be on the conservative side. However, activation of the iron regulon likely occurs as Aft1 is translocated into the nucleolus, so a minimal initial amount of Aft1 (for which we don’t have enough resolution in this system to detect) could be enough for FIT2 and ARN1 induction. By contrast, the Fit2 and Arn1 signal is measuring increase over a background of nothing, so is very easy to detect even at low level induction. To allow the readers to see all our data without over thresholding, we prefer to present the induction of Fit2 and Arn1 at all intensity levels even the very low level induction (green).

(3) "In control strains, expression of Fit2 and Arn1 varied across the population, but generally increased with age": for the right panel, normalization might be more appropriate. What is the fold change in fluorescence during lifespan? Reporting ΔmCherry intensity alone does not provide a quantitative measure of induction.

We have changed the figure to show quantitation as fold change, as suggested.

(4) Figure 6 (model): The model figure is conceptually useful but not easy to follow in its current form; a revised schematic with a clearer depiction of the pathway activations at different replicative ages would be helpful.

We have changed the figure to make the model more clear, as suggested.